# Learning Concept Credible Models for Mitigating Shortcuts

**Jiaxuan Wang**[1]   **Sarah Jabbour**[1]   **Maggie Makar**[1]   **Michael Sjoding**[2]   **Jenna Wiens**[1]

[1]Division of Computer Science & Engineering, University of Michigan, Ann Arbor, MI, USA
[2]Division of Pulmonary and Critical Care, Michigan Medicine, Ann Arbor, MI, USA
Correspondence to: {jiaxuan,wiensj}@umich.edu

## Abstract

During training, models can exploit spurious correlations as shortcuts, resulting in poor generalization performance when shortcuts do not persist. In this work, assuming access to a representation based on domain knowledge (*i.e.*, *known concepts*) that is invariant to shortcuts, we aim to learn robust and accurate models from biased training data. In contrast to previous work, we do not rely solely on known concepts, but allow the model to also learn unknown concepts. We propose two approaches for mitigating shortcuts that incorporate domain knowledge, while accounting for potentially important yet unknown concepts. The first approach is two-staged. After fitting a model using known concepts, it accounts for the residual using unknown concepts. While flexible, we show that this approach is vulnerable when shortcuts are correlated with the unknown concepts. This limitation is addressed by our second approach that extends a recently proposed regularization penalty. Applied to two real-world datasets, we demonstrate that both approaches can successfully mitigate shortcut learning.

## 1   Introduction

In practice, machine learning (ML) models often fail to generalize under distribution shift [1–4], due to shortcut learning [3, 5, 4]. "Shortcut learning occurs when a predictor relies on input features that are easy to represent (*i.e.*, shortcuts) and are predictive of the outcome in the training data, but do not remain predictive when the distribution of inputs changes" [5]. For example, consider building an ML model to predict the severity of knee osteoarthritis from X-ray images [6]. If people with mobility problems in the training set are more likely to have an X-ray using a particular type of mobile X-ray scanner, the model may learn to rely on features related to the scanner type to make a prediction, resulting in a failure to generalize when X-rays are captured from a different scanner.

More formally, consider the causal graph in **Figure 1**, where $Y$ is the target of interest (*e.g.*, diagnosis), $S$ is the shortcut (*e.g.*, scanner type), $X$ is the input (*e.g.*, X-ray image), and $C$ and $U$ are representations that can be inferred from $X$ but are not causally affected by $S$. Here, the dashed bidirectional arrow denotes a spurious correlation that holds during training but not at test time. Solid arrows denote causal relationships that are robust to changes. Note that $S$ only affects the part of the input that is irrelevant for the diagnosis ($X'$), making it causally irrelevant for the prediction. We will consider both settings where $S$ is and is not correlated with $U$. In our example, $C$ could be known radiological risk factors, and $U$ could be unknown radiological risk factors for the disease.

To mitigate a model's reliance on $S$, one can use existing tools if $S$ is observed (*e.g.*, through model interpretation) [5, 7]. However, these methods do not apply when shortcuts are unknown prior to the occurrence of distribution shifts. Moreover, such approaches fail when the spurious correlation is strong (*i.e.*, more convincing shortcuts). In such scenarios, we need additional guardrails. **Our**

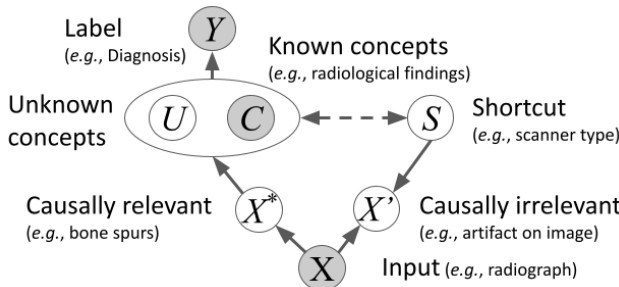

Figure 1: We formalize shortcut learning with a causal graph: $Y$ is the label (*e.g.*, disease diagnosis) and $X$ is the input (*e.g.*, radiograph). $X$ can be decomposed into causally relevant and irrelevant features ($X^*$ and $X'$), *i.e.*, changing $X^*$ changes the label whereas changing $X'$ does not. $X^*$ can be further decomposed into known and unknown relevant concepts ($C$ and $U$). The node surrounding $U$ and $C$ abstracts their interaction (*e.g.*, they can be correlated). A shortcut variable $S$ changes $X'$ and is correlated with $U$ and $C$. Observed variables are in gray. Dashed/solid edges represent correlation that is broken/unaffected under distribution shifts. We aim to eliminate model dependence on $S$.

**approach considers the setting in which we do not have direct knowledge of $S$, but have access to a representation, $C$, that is invariant to $S$** (formally defined in **Section 3.1.1**). It is true that without knowing $S$, we cannot truly confirm whether $C$ is invariant to it. However, in practice, we can rely on established domain knowledge such as risk factors for disease to not encode shortcuts. By exploiting this representation $C$, we mitigate the reliance on shortcuts.

Where does $C$ come from? $C$ arises from domain knowledge and can be elicited in a number of different ways. For example, $C$ may be elicited using transfer learning. Using domain knowledge, experts can identify a related source task. Predictive features (*i.e.*, learned representation) from the related source task (*i.e.*, $C$) can be shared to predict $Y$ [3]. Alternatively, if one has auxiliary concept labels, one can train a model to predict the presence of concepts and use these predictions as $C$ [6]. In fact, both Koh et al. [6] and Jabbour et al. [3] have shown that relying on $C$ alone can outperform a standard model (*i.e.*, a state of the art model) in the presence of shortcuts.

However, depending solely on $C$, referred to as a concept bottleneck model (CBM) [6], ignores potentially unknown concepts (*i.e.*, $U$). When $U$ contains additional useful information, relying solely on $C$ results in inferior predictive performance. E.g., when learning to diagnose the etiology of acute respiratory failure (ARF), clinicians are only accurate about $70\%$ of the time [8–12]. Recently ML models have been shown to improve on this accuracy and have also been shown to generalize across hospitals, suggesting that ML models are using additional features beyond those that humans rely on [13]. We need $U$ because while experts have lots of knowledge about what might be relevant to a specific clinical setting, they certainly do not know everything, or may not be able to always appreciate what is exactly relevant in each setting. For example, Jabbour et al. [13]. found that their model learned that a 'saber sheath trachea' was relevant for identifying patients with a COPD exacerbation. When asking clinicians to come up with the set of features a model should use to identify COPD, most would provide things like 'lung hyperinflation', and miss 'saber sheath trachea,' despite its relevance [14]. More generally, there are likely findings out there that have yet to be recognized as important; new medical discoveries are made all the time. But even when all relevant risk factors are known a priori, sometimes the universe of possibly relevant risk factors is large, such that collecting and labeling all these features for a particular problem is unrealistic. In such cases, the flexibility yielded by including $U$ helps. In some cases, $U$ may turn out to be something known before, *e.g.*, saber sheath trachea, whereas other times it might be a brand new finding.

To tackle the decrease in accuracy when using $C$ alone, we propose two approaches based on *concept credible models* (CCM). The first approach, CCM RES, while simple, is susceptible to a particular failure case when $U$ is correlated with $S$. The second approach, CCM EYE, extends the EYE penalty from Wang et al. [15] to address those issues. The EYE penalty was proposed for linear models as a way to increase the model's alignment with expert knowledge (in our case, $C$) without sacrificing predictive performance. Here, we hypothesize that the same idea can mitigate the use of shortcuts. We thus extend the EYE penalty to work with non-linear models by applying it to the learned representation/concept space. This is a nontrivial application of the EYE penalty since here

the concept space is not equivalent to the input space. Moreover, we identify the conditions in which CCM RES and CCM EYE mitigate learning shortcuts. In summary, our contributions include

- We propose the idea of learning concept credible models (CCM), in which $C$ is not required to be directly represented in the input space. We demonstrate that CCMs are more robust to shortcuts compared to existing approaches.

- Unlike previous work on shortcut learning, we show that our approaches still apply when shortcuts are *perfectly* correlated with other features of $C$, and address the limitation of existing methods that rely solely on $C$ in making predictions.

- Theoretically, we identify the sufficient conditions under which a CCM can eliminate shortcuts for the setting considered in **Figure 1**.

- Empirically, we demonstrate that our approach can still help mitigate shortcuts even when these conditions identified above are violated.

## 2    Related work

The idea of "concept credible models" is connected to multiple fields in ML, explained below.

**Connection to shortcut learning**. Shortcut learning is a particular failure mode that arises due to distribution shifts [2, 16]. However, to date, researchers have typically assumed shortcuts are known *a priori*. Under such settings, one can augment the dataset to decorrelate shortcuts with data [17–21] or regularize model parameters to not rely on shortcuts [7, 5, 22]. In contrast, we do not assume that we know $S$. This change makes approaches such as IRM [16] and REx [23] no longer applicable, because without knowing $S$, it is hard to specify the family of distributions to which a model should be robust. The above methods also will not work when shortcuts and robust features are perfectly correlated because without prior knowledge, they cannot be separated apart.

**Connection to concept bottleneck models**. The concept bottleneck model (CBM) was proposed in Koh et al. [6] with the goal of making a model's decision more transparent by only using $C$ for prediction. While this can mitigate shortcuts since the model is forced to rely on $C$ instead of spurious correlations, it ignores unknown concepts, often resulting in lower accuracy compared to a standard model [24]. We address this problem by adding a channel that takes $X$ as input, in addition to predicting $Y$ from $C$. This added channel, along with a carefully chosen regularizer, enable CCM to learn $U$, resulting in better accuracy. We note that the added channel design also appears in a concurrent work [25]. However, unlike CCM, [25] does not have a mechanism to mitigate learning shortcuts. Instead, it makes the added concepts interpretable, which is complementary to our work.

**Connection to credible learning**. Credible models are trained by regularizing a model's feature attribution to be close to expert identified features (*i.e.*, features known to be relevant for the prediction), in addition to being accurate [15, 26]. While credible learning has been shown to work well in a transfer learning setup within natural language processing [26], we are the first to study its applicability to mitigate the effects of shortcut learning. However, unlike previous work, we do not require domain knowledge (*i.e.*, concepts) to be expressed directly in the input space. This provides us with greater flexibility in exploring different types of inputs in which it may be difficult to collect domain expertise (*e.g.*, the pixel value of images). As a result, our approach does not require models to be linear.

## 3    Proposed approach

We formalize the problem and assumptions, and propose methods to learn concept credible models.

### 3.1    Preliminaries

To simplify the exposition, we illustrate the setup for a regression problem. The setting, however, is easily adaptable to multi-class classification. We capitalize random variables and bold vectors. For example, $\boldsymbol{x}$ denotes an instance of the random vector $X$. We denote the Pearson correlation between two random variables as $corr(\cdot, \cdot)$.

### 3.1.1    Setup & assumptions

$Y$ is the target of interest (*e.g.*, diagnosis), $S$ is the shortcut (*e.g.*, scanner type), $X$ is the input (*e.g.*, X-ray image). $C$ and $U$ are representations that can be inferred from $X$ but are not causally affected by $S$.

Given a dataset $\mathcal{D} = \{(\boldsymbol{x}^{(i)} \in \mathbb{R}^d, y^{(i)} \in \mathbb{R})\}_{i=1}^n$ of $n$ samples generated according to **Figure 1** and a function $f_c : \mathbb{R}^d \to \mathbb{R}^c$ such that $C := f_c(X)$, **we aim to learn an accurate prediction from the input variable $X$ to the target variable $Y$ ($f : \mathbb{R}^d \to \mathbb{R}$) that is invariant to the *unknown* shortcut $S$.** Here, $c$ and $d$ are dimensionality for $C$ and $X$ respectively. We also assume that there exists *unknown* concepts $U := f_u(X) \in \mathbb{R}^u$ that are invariant to $S$ with dimensionality $u$.

Although we assumed that $f_c$ is given in this setup, we can also learn it from a related dataset (*e.g.*, predictive features for this related task). Note that this does not require direct knowledge of $S$. E.g., it is sufficient to know that the shortcut for the target task is unlikely a shortcut for the related task.

In this paper, invariance refers to counterfactual invariance from Veitch et al. [22] (**Definition 1.1**). Adapting their notation, let $X(s)$ denote the counterfactual $X$ we would have seen had $S$ been set to $s$, leaving all else fixed, $f$ is *counterfactually invariant* to $S$ if $f(X(s)) = f(X(s'))$ almost everywhere, for all $s, s'$ in the sample space of $S$. This invariance ensures generalization of the model regardless of the shortcut's distribution. Note that invariance is not the same as independence. E.g., scanner type does not cause the diagnosis (*i.e.*, diagnosis is invariant to scanner type) yet they can be correlated.

We require two assumptions about $C$. **A1** is implied by the causal graph, while **A2** is not.

**A1: $C$ is counterfactually invariant to $S$.** *E.g.*, changing the scanner type does not change the occurrence of a bone spur in an X-ray image. Thus the presence of bone spur is invariant to the scanner type. Without this assumption, even a model that only uses $C$ may indirectly depend on $S$.

**A2: $S$ is redundant given $C$ (*i.e.*, $Y \perp\!\!\!\perp S|C$).** *E.g.*, given bone spur and other radiological findings from an X-ray image, the type of scanner is irrelevant in predicting arthritis severity. Without this assumption, including $S$ improves accuracy.

Since both **A1** and **A2** are not testable without knowing $S$, in experiments, we test our methods' sensitivity to each assumption empirically. Note that we do not make any assumption regarding the correlation between $S$ and $U$.

### 3.1.2 Existing approaches

We formally introduce common methods that are typically used in this prediction setting.

**Standard Model**: Standard model refers to the task specific state-of-the-art model trained with loss $L$ with empirical risk minimization: $\arg\min_{f_{\text{STD}}} \sum_{i \in [1, \cdots, n]} L(f_{\text{STD}}(\boldsymbol{x}^{(i)}), y^{(i)})$. Such a model cannot distinguish among $C, U$, and $S$. Thus, the model is vulnerable to relying on shortcuts for prediction.

**Concept bottleneck model (CBM)**: Our proposed approach builds on CBM [6]. CBM's prediction can be written as $f_{\text{CBM}}(X) = f_y(f_c(X))$. Here, $f_y$ maps from $C$ to $Y$ and is trained using empirical risk minimization: $\arg\min_{f_y} \sum_{i \in [1, \cdots, n]} L(f_{\text{CBM}}(\boldsymbol{x}^{(i)}), y^{(i)})$. When $U$ contains additional information useful in predicting $Y$ given $C$, CBM is less accurate than a standard model.

### 3.1.3 A motivating example

To build intuition, consider the following linear regression example in which $C$ and $S$ are perfectly correlated during training (*i.e.*, $C = S$ in $\mathcal{D}$) while $C$ and $U$ are not. Given $X = [C, S, U]$ and $Y = C + U$, a least squares linear regression solution gives a prediction of $\hat{Y} = (1 - t)C + U + tS$ (derived in A.2). The free parameter $t \in \mathbb{R}$ results from the spurious correlation between $C$ and $S$.

The minimum $L_2$ norm solution of this problem results in $t = 0.5$ and will fail to generalize when the correlation between $S$ and $C$ no longer holds at test time. In contrast, if we only use $C$ for prediction (*i.e.*, CBM), the solution will not achieve a loss of 0 since it ignores $U$. Furthermore, in cases where $C$ and $U$ are correlated, CBM is asymptotically biased due to omitting the variable $U$ [27]. This means that models that ignore $U$, such as CBM, cannot recover the true regression coefficients even as the training set size approaches infinity.

## 3.2 Proposed approaches: concept credible models

We introduce two approaches to learn a concept credible model with the goal of mitigating shortcuts: CCM RES and CCM EYE.

### 3.2.1 CCM RES

The limitation of CBM stems from its inability to infer $U$ from $X$. To address this limitation, we design a two stage approach, CCM RES, that first fits a CBM on the dataset, and then fits a residual model $f_x$ based on the difference between $Y$ and the output of the CBM. This idea is similar in spirit to boosting methods. $f_x$ enables CCM RES to learn $U$. CCM RES obtains its prediction by adding the output from $f_x$ to the output of the CBM: $f_{\text{RES}}(\boldsymbol{x}) = f_{\text{CBM}}(\boldsymbol{x}) + f_x(\boldsymbol{x})$. When CBM achieves small training loss (*e.g.*, the difference between $Y$ and CBM's prediction is small), CCM RES does not have to rely on information other than $C$. Otherwise, CCM RES relies on $f_x$ to make up for what $C$ alone cannot learn. We learn CCM RES with empirical risk minimization: $\hat{f}_{\text{RES}} = \arg\min_{f_x} \sum_{i \in [1, \cdots, n]} L(f_{\text{RES}}(\boldsymbol{x}^{(i)}), y^{(i)})$.

Applied to **Example 3.1.3**, when $U$ is independent from $S$ in $\mathcal{D}$, the resulting model not only achieves $0$ empirical loss, but also has $0$ reliance on $S$, achieving our goal of accuracy without relying on shortcuts. We generalize this motivating example to all linear models below.

To highlight the strength of CCM RES compared to previous approaches, we consider the worst case scenario where $|corr(C, S)| = 1$. In this case, the system is under-specified (*i.e.*, allowing multiple minimum loss solutions), and a standard model may not be consistent with the causal DAG. Previous approaches fail in such scenarios since they cannot distinguish $C$ from $S$.

**Consistency of CCM RES.** If $f_{\text{RES}}$ is linear (both $f_{\text{CBM}}$ and $f_x$ are linear), $X = [C, S, U]$, $Y = aC + bU + \epsilon$ with $a, b \in \mathbb{R}$ and a zero mean error $\epsilon$, $|corr(C, S)| = 1$ on the training distribution $P_{XY}$, and $U \perp\!\!\!\perp S$, $\arg\min_{f_{\text{RES}}} \mathbb{E}_{x, y \sim P_{XY}} (y - f_{\text{RES}}(x))^2$ recovers the true parameters without relying on $S$ (*i.e.*, weight $a$ for $C$, $b$ for $U$, and $0$ for $S$).
*Proof.* We first show that the residual is independent of $S$, thus fitting to the residual will not use $S$. From $U \perp\!\!\!\perp S$ and $|corr(C, S)| = 1$, we know $U \perp\!\!\!\perp C$. Fitting on infinite data with squared loss simplifies $f_{\text{CBM}}$ to $\mathbb{E}(Y|C) = \mathbb{E}(aC + bU + \epsilon|C) = aC + b\mathbb{E}(U|C) = aC + b\mathbb{E}(U)$. The residual, $Y - \mathbb{E}(Y|C) = b(U - \mathbb{E}(U)) + \epsilon$, is independent of $S$ because $U \perp\!\!\!\perp S$. Thus the prediction is $aC + bU$, recovering the true parameters. $\qquad\qquad\square$

**Remark**: As the $|corr(S, C)|$ decreases, the system may no longer be under-specified, and both CCM RES and a standard model will be consistent. This happens when $S$ is not a linear combination of $C$ and $U$, in which case the minimum loss solution is unique. If, however, $S$ is a function of $U$, we cannot distinguish $S$ from $U$ and thus cannot guarantee the consistency of CCM RES.

This result shows that a linear CCM RES, unlike a linear CBM, is a consistent estimator. However, while CCM RES enables learning unknown concepts, it fails when $S$ and $U$ are correlated because the residual can be estimated as a linear combination of $U$ and $S$, making $f_x$ vulnerable to encoding a shortcut. We address this problem with a second approach.

### 3.2.2 CCM EYE

In our second approach, we utilize the EYE regularization from Wang et al. [15] to learn a concept credible model (CCM EYE). The EYE penalty penalizes reliance on features that are correlated with $C$ but not in $C$. We propose to apply EYE regularization on the concept space (*i.e.*, the learned representation space) as follows: $f_{\text{EYE}}(\boldsymbol{x}) = \boldsymbol{\theta}_x^\mathsf{T} f_x(\boldsymbol{x}) + \boldsymbol{\theta}_c^\mathsf{T} f_c(\boldsymbol{x})$ where $f_c$ computes the known relevant representation $C$ and $f_x$ computes a representation from the last layer of a standard model. This transformation from $\boldsymbol{x}$ to $f_x(\boldsymbol{x})$ allows the model to be non-linear. $\boldsymbol{\theta}_x$ and $\boldsymbol{\theta}_c$ are coefficients for $f_x(\boldsymbol{x})$ and $f_c(\boldsymbol{x})$ respectively. We then apply the EYE regularization on those parameters:

$$\hat{f}_{\text{EYE}} = \arg\min_{\boldsymbol{\theta}_x, \boldsymbol{\theta}_c} \sum_{i \in [1, \cdots, n]} L(f_{\text{EYE}}(\boldsymbol{x}^{(i)}), y^{(i)}) + \lambda J([\boldsymbol{\theta}_x, \boldsymbol{\theta}_c]) \tag{1}$$

Here, $J([\boldsymbol{\theta}_x, \boldsymbol{\theta}_c]) = \|\boldsymbol{\theta_x}\|_1 + \sqrt{\|\boldsymbol{\theta_x}\|_1^2 + \|\boldsymbol{\theta_c}\|_2^2}$ is the EYE regularization applied to our setting and $\lambda \in \mathbb{R}_{\geq 0}$ is a hyperparameter that controls the trade-off between regularization and loss. The EYE penalty more strictly penalizes $\boldsymbol{\theta_x}$ compared to $\boldsymbol{\theta_c}$, allowing the norm of $\boldsymbol{\theta_c}$ to be larger and hence encouraging the model to rely on $C$. Conversely, $J$ discourages the use of $X$, which include both $U$ and $S$. If $U$ is in fact important in predicting $Y$, the minimization of the loss encourages the use of $U$ more than $S$ because of **A2**: $U$ has more predictive power compared to $S$ given $C$.

We choose $\lambda$ such that $\boldsymbol{\theta_x}$ is strictly regularized without sacrificing in-distribution performance (*i.e.*, performance under the biased training distribution). We do so by picking the largest $\lambda$ such that the model's accuracy on the validation set is not statistically worse than that of a standard model, where statistically worse performance is measured in terms of empirical $95\%$ bootstrapped confidence intervals. This ensures that CCM EYE maximizes the use of $C$ without sacrificing predictive performance.

Similar to CCM RES, CCM EYE is a consistent estimator even when $|corr(C, S)| = 1$. The remark for CCM RES applies to CCM EYE as well.

**Consistency of CCM EYE.** If $f_{\text{EYE}}$ is linear, $X = [C, S, U]$, $Y = aC + bU + \epsilon$ with $a, b \in \mathbb{R}$ and a zero mean error $\epsilon$, $|corr(C, U)| \neq 1$ and $|corr(C, S)| = 1$ on the training distribution $P_{XY}$, $\arg \min_{f_{\text{EYE}}} \mathbb{E}_{x, y \sim P_{XY}} (y - f_{\text{EYE}}(x))^2 + \lambda J([\boldsymbol{\theta_x}, \boldsymbol{\theta_c}])$ recovers the true parameters (*i.e.*, weight $a$ for $C$, $b$ for $U$, and $0$ for $S$) with standardized input, where $\lambda$ is chosen as described before from $P_{X, Y}$.

*Proof.* With infinite data, the empirical loss is the in-distribution generalization loss, therefore $\lambda$ is chosen such that the generalization loss is minimized. In the worst case scenario, the perfect correlation between $S$ and $C$ makes this linear problem underspecified (*i.e.*, multiple solutions), which means $\lambda$ is non-zero because it is set to be the largest value such that model performance is not statistically worse than a standard model trained on $P_{X, Y}$. Fixing the same loss, the EYE penalty places zero weight on standardized features (features normalized to zero mean and unit variance) that are perfectly correlated with expert identified features [15]. Treating $C$ as the expert identified feature, the coefficient for $S$ is thus $0$. Combined with the fact that $C$ and $U$ are not perfectly correlated, to achieve the minimum loss, the coefficients for $C$ and $U$ must be $a$ and $b$ respectively. $\qquad\square$

**Remark**: Unlike CCM RES, the consistency of CCM EYE does not require $U \perp\!\!\!\perp S$. Intuitively, EYE can separate $U$ from $S$ because of **A2**: $U$ is needed in addition to $C$ to be accurate, yet $S$ is not needed given $C$. Note that $|corr(S, C)| = 1$ does not imply $U \perp\!\!\!\perp S$ because $U$ can be correlated with $C$, which in turn is correlated with $S$.

Our consistency results are *not* limited to the regression setting. One can replace the mean squared loss in the consistency proof with the logistic loss or the negative log likelihood. Under those losses, the same results hold and the proposed approaches do not rely on $S$.

For our theoretical results, we focused on the linear case. In this setting, perfect correlation is required to create model underspecification. When features are linearly independent, all methods are consistent. Encouraged by experiments with non-linear models (Section 4), we believe these results could be further expanded upon (e.g., non-linear setting) in future work. While our theoretical results on linear models are restrictive, they serve as a sanity check and hint at what we might expect with more complex models as their last layers are often linear (*e.g.,* neural networks). We also note that the additive structure of both CCM RES and CCM EYE does not restrict their expressive power as $f_x$ can be arbitrarily complex, capturing the interactions between $C$ and $U$.

## 4   Experiments & results

In this section, we verify CCM's robustness to spurious correlations on three classification tasks using publicly available datasets. The first is an image classification task similar to the one examined by Koh et al. [6]. This task demonstrates the superior performance of CCM when $C$ is complex and non-linear in $X$. The second task is the prediction of pulmonary edema from chest radiographs. It demonstrates CCM's effectiveness in a critical domain where accuracy and robustness are needed. We include an additional task to predict in-hospital mortality in Appendix A.6. To demonstrate that the shortcuts used in our experiments are not easily observable, we include results of feature attribution of the baselines as well as our proposed approaches on both image datasets in Appendix A.7.

**Concept Acquisition:** In this paper, we explored two practical settings to obtain $C$, one using auxiliary labels (the bird attributes) and another using a transfer learning setup (transfer from the cardiomegaly task to the edema task). In both settings, we show that our approaches are effective at mitigating shortcuts. While there are more ways to obtain $C$, we defer a comparison of the benefits/tradeoffs of various approaches to future work.

Note that we do not compare the approaches on common computer vision datasets such as CIFAR 10/100 [28] because $C$ is not readily defined. Instead, we relied on benchmark datasets that have been

used in past work involving $C$ [6]. We also note that in contrast with the theoretical results, which focused on linear models, the experiments do not make those assumptions. For example, the edema experiment uses sex as the shortcut, which is not a linear concept in terms of the input (*e.g.*, pixels)[1].

We start with a setting in which our assumptions hold (**Section 3.1.1**), and then relax our assumptions to stress test our methods. We evaluate on both biased and unbiased/clean data (defined in the evaluation section of each task) to explore the effects of a distribution shift caused by the shortcut. All models are trained and selected using only the biased dataset (*e.g.*, the dataset where $corr(S, Y) \neq 0$).

**Baselines.** We compare CCM with the following methods:

- **STD($X$)** is a model trained end-to-end on the biased dataset [29], using $X$ to predict $Y$. We expect it to learn $S$ because there is a backdoor path from $S$ to $Y$ in **Figure 1**.

- **Concept bottleneck model (CBM)** removes STD($X$)'s reliance on $S$ by only fitting on $C$ [6]. However, CBM lacks the ability to infer $U$ and thus may sacrifice discriminative performance before and after shortcut induced distribution shifts. Following Koh et al. [6], we train a CBM by fitting a logistic regression model on top of $C$.

- **STD($C$, $X$)** is a standard model that conditions both on $X$ and $C$ for prediction. On the one hand, we expect this baseline to be more robust than STD($X$) when $S$ breaks because it has an easy access to $C$. On the other hand, conditioning on $X$ gives the baseline the ability to infer $U$, unlike CBM. However, when $S$ is highly correlated with $C$, this baseline can still rely on $S$ to make a prediction. While there are many ways to implement STD($C$, $X$), we implement this baseline as a special case of CCM EYE with $\lambda = 0$. This allows us to clearly demonstrate the effect of EYE regularization on model robustness.

## 4.1 Experiments on the CUB dataset

The **Caltech-UCSD Birds-200-2011** dataset (CUB) consists of $11,788$ images of birds [30] each belonging to one of $200$ species ($Y$). In addition to the images, the dataset also contains $312$ binary attributes/concepts (*e.g.*, beak color) describing birds in each image. Following Koh et al. [6], we filter out concepts with noisy annotations.

**Concept Acquisition:** We train a standard model to predict those concepts from $X$, with random Gaussian noise $N(0, 0.1^2)$ added to the image. The resulting prediction is treated as $C$. Note that no shortcut is introduced in obtaining $C$, in order to satisfy **A1**. Later, we will break this assumption.

**Shortcut:** The shortcut we consider here is the level of noise, $\sigma$, in an image. We correlate $\sigma$ with bird species to mimic a setting where some birds have noisier photos than others because they are harder to observe in the wild. Ideally, a model should be able to classify the birds regardless of the noise level. To introduce $\sigma$ as a shortcut, we correlate it with bird classes in the training data. However, we do not correlate $\sigma$ with $Y$ directly because it violates **A2** (a setting we explore in Appendix A.4). Instead, we correlate $\sigma$ to $Y$ through $C$, using the BIAS function described in Appendix A.3.

| Method | Test Acc (biased) | Test Acc (clean) |
|---|---|---|
| CCM EYE | 76.0 (75.0, 77.2) | 75.2 (74.1, 76.5) |
| CCM RES | 75.6 (74.2, 76.9) | 76.0 (74.8, 77.2) |
| STD(X) | 75.7 (74.7, 77.0) | 55.8 (54.7, 57.3) |
| CBM | 71.6 (70.4, 72.9) | 72.8 (71.7, 73.9) |
| STD(C,X) | 76.0 (74.7, 77.2) | 69.7 (68.6, 70.8) |

Table 1: On the CUB dataset, when **A1** and **A2** hold, CCM is no worse than baselines on the biased data (column 1), and is better than baselines on the clean data (column 2). Empirical $95\%$ CI are in parentheses.

**Model Training**: Following [6], all methods use an Inception V3 architecture [29] initialized using the Imagenet dataset [31]. We divide the training set predefined in the CUB dataset into train and validation set with a $80/20$ random split, and use the predefined test set for evaluation. We report the performance on this test set as the result for unbiased/clean dataset. We then add class dependent noise described earlier to the train, validation and the test set to form the biased dataset. All methods are trained on the training set of this biased dataset using SGD with learning rate of $0.01$, momentum of $0.9$, and batch size of $32$. We apply $10^{-4}$ weight decay to each model and decay the learning rate every 15 epochs. For CCM EYE, we tune $\lambda$ in the range of $[10^{-2}, 10^{-3}, 10^{-4}, 10^{-5}, 10^{-6}]$.

---

[1]Code is available at `https://gitlab.eecs.umich.edu/mld3/ConceptCredibleModel`

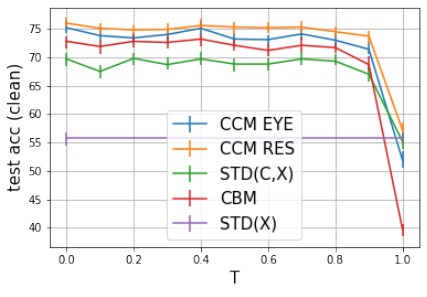 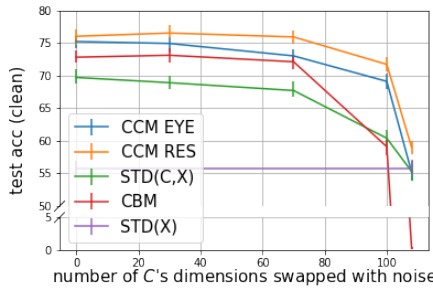

(a) Test Accuracy (clean) violating **A1**  (b) Test Accuracy (clean) violating **A2**

Figure 2: **(a)** Model performance under the clean test set when violating **A1**. When $C$ is learned using a biased dataset (sweeping $T$ on the horizontal axis), we violate **A1**. Unless $C$ is extremely corrupted (*e.g.*, $T = 1$), CCM performs relatively well. **(b)** Model performance under the clean test set when violating **A2**. $C$ becomes less informative when replaced with noise, presenting an advantage to using $S$ and violating assumption **A2**. Despite this, CCM still performs well, even when large portions of $C$ are irrelevant for the prediction.

**Evaluation**: Recall that our goal is to learn an accurate model without using $S$. We generate the biased and the clean test set as described in the model training section above. Evaluating on the biased test set demonstrates how the model performs when $S$ is correlated with $Y$. Evaluating on the clean test set demonstrates how the model performs without image noise (*i.e.*, the shortcut no longer exists). A model that does not rely on $S$ should perform similarly on both datasets. We measure performance on the CUB dataset using accuracy (ACC) as bird classes are balanced. Empirical $95\%$ confidence intervals are reported based on bootstrapped samples from the test set.

**Results**: We examine the results when **A1** and **A2** hold/break. We also explore varying $\lambda$ in Appendix A.4 to justify its choice.

Q1: How does CCM perform when **A1** and **A2** are satisfied?

Both CCM RES and CCM EYE are no worse than baselines when tested on the biased dataset (**Table 1** first column), but are significantly better than baselines when $S$ is removed (**Table 1** second column). In contrast, STD($X$) performs well on the biased dataset but underperforms on the clean dataset, indicating its reliance on $S$. As expected, CBM does not rely on the shortcut as its performance is stable with and without $S$. However, its inability to utilize $U$ results in a drop in accuracy compared to others. Finally, STD($C, X$) is accurate on the biased dataset and improves over STD($X$) on the clean dataset because it encourages the model to use $C$. However, CCM EYE dominates, suggesting that conditioning on both $C$ and $X$ is not enough to remove model reliance on $S$.

Q2: How does CCM perform when **A1** is violated?

We relax **A1** by learning $C$ on a biased dataset. Specifically, we use the BIAS function introduced in **Section A.3** but vary the probability of correlating $S$ with $Y$ (*i.e.*, varying the parameter $T$). **Figure 2a** shows the results of varying $T$ on the test accuracy in the clean data. As before, CCM dominates the other baselines except at $T = 1$, indicating that unless $C$ is extremely corrupted with $S$, CCM performs well compared to the other models. We note that STD($X$) does not change with $T$ because it does not use $C$. In Appendix A.4, we show that all methods except CBM perform similarly well on the biased dataset.

Q3: How does CCM perform when **A2** is violated?

We relax **A2** in two ways: a) $S$ contains information outside of $C$ but within $U$ (*i.e.*, $Y \perp\!\!\!\perp S | C, U$), and b) $S$ contains information outside of $C$ and $U$ (*i.e.*, $Y \not\perp\!\!\!\perp S | C, U$).

To violate **A2**, $S$ can no longer be redundant given $C$. To achieve this, we first introduce a correlation between $S$ and $C$ as before (to satisfy **A1**) and then we randomly replace columns in $C$ with Gaussian noise $N(0, 1)$, but keep $S$ the same. This procedure correlate $S$ with $U$ because the swapped out information becomes unknown concepts based on which $S$ is generated. The more concepts swapped

for noise, the less informative $C$ becomes, increasing the relative value of $S$ in predicting $Y$. For example, when 100 random dimensions of $C$ are replaced with noise, a linear model trained with ($C$, $S$) significantly outperforms a model based on just $C$.

This concept swapping greatly affects CBM because it relies solely on $C$, which is corrupted. When $S$ is removed (**Figure 2b**), both CCM EYE and CCM RES outperform baselines until all concepts are corrupted (in which case CCM performs similarly to a standard model). The performance of $STD(X)$ is constant across settings because it does not rely on $C$. This experiment also shows that not all dimensions of $C$ need to be relevant to the prediction for CCM to work. This is a desirable property as **expert knowledge with respect to relevant concepts could be flawed**.

In the case where $S$ contains information outside of $C$ and $U$ (*i.e.*, $Y \not\perp\!\!\!\perp S|C, U$), our findings are similar (Appendix A.4). All methods except CBM perform well on the biased dataset.

## 4.2 Experiments on the MIMIC dataset

The **MIMIC-CXR** dataset [32, 33] consists of chest X-rays and corresponding radiology reports. These data can be linked to MIMIC-IV [34, 33], which contains de-identified clinical data. Each chest X-ray is associated with 14 text-mined radiology report labels corresponding to 14 different radiological findings. Among these findings, we aim to predict a diagnosis of edema (excess fluid in the lungs).

**Concept Acquisition:** Oftentimes in healthcare, there exist related tasks from on which one can draw concepts. Thus, here, we explore a transfer learning setup to extract concepts. Based on domain knowledge, we chose cardiomegaly (enlarged heart) as the source task to learn $C$. Patients with cardiac dysfunction are more likely to develop heart failure, and pulmonary edema can develop as a consequence of heart failure [35, 3]. Thus, we expect that predictive features of cardiomegaly are useful concepts in diagnosing edema. After excluding images without labels the cardiomegaly/edema tasks contained $108, 785/107, 510$ X-rays, respectively. To obtain $C$, we trained an Inception V3 network pretrained on the ImageNet dataset to predict cardiomegaly. Then we used the last layer representation of the network as $C$ (dimension $2048$).

**Shortcut:** We introduce a realistic shortcut based on patient sex. We increased the correlation between male and edema by dropping $T$ proportion of females/males with/without a positive label. Prior to resampling, male was only mildly correlated with cardiomegaly (corr. coefficient of $-0.025$; Empirical $95\%$ bootstrapped CI of $(-0.031, -0.019)$).

**Model Training**: Similar to the CUB experiment, we used the Inception V3 network initialized using the ImageNet dataset as the prediction model. We divided the chest X-ray datasets into train, validation, and test sets with a 64/16/20 random split. Then, we resampled the edema dataset such that male and edema are correlated. All methods were trained on this biased edema dataset. The hyperparameter search range was the same as the CUB experiments.

**Evaluation**: Since both $S$ and $Y$ are binary, we can resample the test set to vary the correlation between $S$ and $Y$ to stress test our model under different testing distributions. In particular, we swept the correlation between sex and edema from $-1$ (reversing the training correlation) to $1$ (extremely biased distribution). A model robust to the sex shortcut should do well in all settings.

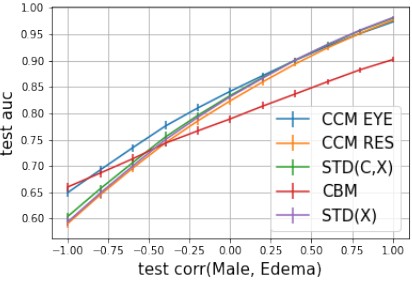

Figure 3: Result of the MIMIC-CXR experiment. The model is trained on a biased dataset where $S$ and $Y$ has a correlation of $0.65$ and tested on subsampled dataset with different correlation. The result shows that CCM EYE is consistently better than baselines. The error bars are the $95\%$ confidence intervals bootstrapped on the test set.

**Results**: The performance of all methods is significantly affected when the test correlation is decreased to the point of reversal with the correlation in the training set (**Figure 3**). This is inevitable as the shortcut provides information to predict $Y$ given $C$, violating **A2**. However, across the range

of test correlation settings, when trained on the biased distribution (correlation between male and edema is $0.65$), CCM EYE performs consistently better than baselines. Compared to CBM, CCM EYE is more effective when the testing correlation is similar to the training correlation. Compared to other baselines, CCM EYE is most effective when the shortcut is negatively correlated with the outcome, demonstrating the robustness of CCM EYE against the sex shortcut. Similar trends hold when we vary the training correlation between male and edema (Appendix A.5).

## 5  Discussion & conclusion

In this work, we proposed two approaches that use domain knowledge $C$ to learn an accurate model while mitigating the use of shortcuts. Our methods do not assume $C$ to be sufficient to make an accurate prediction and apply even to scenarios where $|corr(S, C)| = 1$, settings previous work have not addressed. Between our proposed methods, we recommend using CCM EYE as a) it does not rely on $U \perp\!\!\!\perp S$, and b) empirically it outperformed CCM RES on the clinical datasets. Applied to two datasets, we show that CCMs successfully reduce shortcut learning without sacrificing accuracy, even when our assumptions that $C$ is invariant to $S$ and $S$ is redundant given $C$ do not hold.

Our work is not without limitation. First, we rely on the ability to obtain a good $C$. In our experiments, we explore two different approaches to extract $C$, with some success. However, there may be other approaches that apply as well. Second, other regularizers may further increase credibility (*e.g.*, $\|\boldsymbol{\theta}_x\|_2^2$). Future work could consider the task of finding the optimal credible regularizer to mitigate using shortcuts. Third, we did not explore grounding/interpreting the implicitly learned features ($U$), which is important for the practical use of CCM. Indeed, while CCM is capable of ruling out shortcuts that are redundant given $C$, it does not replace the need to carefully interpret and examine concepts picked up by the model (*i.e.*, $U$), as some shortcuts contain more information given $C$. This requires examining what $U$ encodes and having domain experts validate the learned concepts. Fortunately, this is an active area of research and there are already ways to close the interpretation gap (*i.e.*, interpret learned concepts), both supervised [36] and unsupervised [37, 25]. Complementing CCM with the interpretation of $U$ is an interesting future direction that could work to further mitigate shortcut learning. Nonetheless, we expect CCM to be a step towards building trustworthy systems that can be safely applied in practice.

## 6  Acknowledgments

This work and the dissemination of this work was supported by the National Science Foundation (NSF; award IIS-1553146 and grant 2153083) and the National Heart Lung and Blood Institute of the National Institutes of Health (NHLBI; grant R01HL158626). The views and conclusions in this document are those of the authors and should not be interpreted as necessarily representing the official policies, either expressed or implied, of the National Science Foundation, nor of the National Institutes of Health.

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
