# A Appendix

## A.1 Additional Discussion

**Ethical Considerations and Societal Impact.** In general, algorithms that are developed for high-stakes domains, such as healthcare, should be carefully validated before deployment into real-world settings. In our work, we consider the application of diagnosing disease based on chest X-rays, with the goal of preventing a model from using a shortcut, $S$ present in biased training data. The approach assumes that $C$ is invariant to $S$. While one may test the robustness of our approach in preventing the use of many measurable shortcuts, such as patient age, sex, and race, there exists no exhaustive list of such shortcuts. Thus, it is important to thoroughly validate such algorithms.

## A.2 Derivation of the least squared solution for Section 3.1.3

Given $X = [C, S, U]$, $Y = C + U$, $corr(C, U) \neq 1$, and $C = S$ in $\mathcal{D}$, a least squares linear regression solution gives a prediction of $\hat{Y} = (1 - t)C + U + tS$.
*Proof.* We know $C + U$ is a solution because they give $0$ loss. Since $C = S$ in the dataset, $(1 - t)C + tS + U$ is also a solution for any $t \in \mathbb{R}$. Since $U$ and $C$ are not co-linear, the solution has rank 2. By the rank nullity theorem, we know the null space has dimension 1 (because the dimension of input is 3), thus its least squares solutions also has dimension of 1, which shows that $(1 - t)C + tS + U$ are all the solutions that minimizes the loss. $\square$

The minimum $L_2$ norm solution of this problem results in $t = 0.5$
*Proof.* Given the solution is $(1-t)C+tS+U$, we minimize the coefficient with $L_2$ loss: $\arg\min_t (1-t)^2 + t^2 + 1$, which solves to $t = 0.5$. $\square$

If we only use $C$ for prediction (*i.e.*, CBM), the solution will not achieve a loss of $0$ since it ignores $U$.

*Proof.* Even with infinite training data, fitting $Y$ using $C$ results in $\mathbb{E}(Y|C) = C + \mathbb{E}(U|C)$. The $L_2$ loss with $Y$ is thus $\mathbb{E}((U - \mathbb{E}(U|C))^2)$, which is non-zero when $C$ and $U$ are not co-linear. $\square$

## A.3 Introducing a shortcut in CUB dataset

To introduce $\sigma$ as a shortcut, we correlate it with bird classes in the training dataset. Specifically, we correlate $\sigma$ to $Y$ through $C$, using the following function:

```
1: function BIAS(X)
2:     ss ← linspace(0, 0.1, n_σ)
3:     if Uniform(0, 1) < T then
4:         σ ← ss[arg max(CBM_O(X))  mod n_σ]
5:     else
6:         σ ← randomChoice(ss)
7:     end if
8:     return  X + N(0, σ²)
```

The algorithm starts by uniformly spacing $n_\sigma$ levels of noise between 0 and 0.1 (line 2). Setting $n_\sigma$ to 200 would be equivalent to each bird species having its own level of noise. To simulate a more realistic setting, we arbitrarily set $n_\sigma = 10$, allowing multiple birds to share a noise level. We show in A.4 that varying $n_\sigma$ does not affect our results. We then use modular arithmetic to ensure that each of the 200 bird types gets mapped onto one of the 10 noise levels. Then with probability $T$, we correlate $\sigma$ with the bird class predicted from an oracle model $CBM_O$. $CBM_O$ is similar to the CBM baseline explained above, with the exception that it has oracle access to a noise free dataset at training time (line 3-4) [2]. By using predictions from $CBM_O$ rather than the true labels, we ensure that $\sigma$ contains information about $Y$ through $C$. We test breaking this assumption later. Then with probability $1 - T$, we break the correlation between $\sigma$ and $Y$ by randomly choosing a noise level (line 5-6). Finally we return an image with shortcut.

---

[2]We note that $CBM_O$ is not a valid baseline as it is trained on the clean dataset, while all baselines are trained on the biased dataset. It simply serves as an approach to satisfy **A2**.

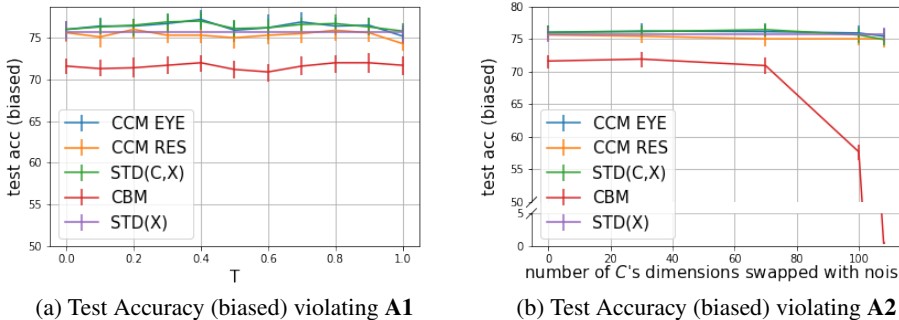

(a) Test Accuracy (biased) violating **A1**  (b) Test Accuracy (biased) violating **A2**

Figure 4: **(a)** When **A1** is broken by adding bias to how $C$ is trained, the biased dataset performances are constant across methods. Note that except for CBM, all methods performed about the same. **(b)** When **A2** is broken by replacing dimensions of $C$ with random noise, the predictive power of CBM decreases, yet other methods have similar performance on the biased dataset because they can learn from $X$ in addition to $C$.

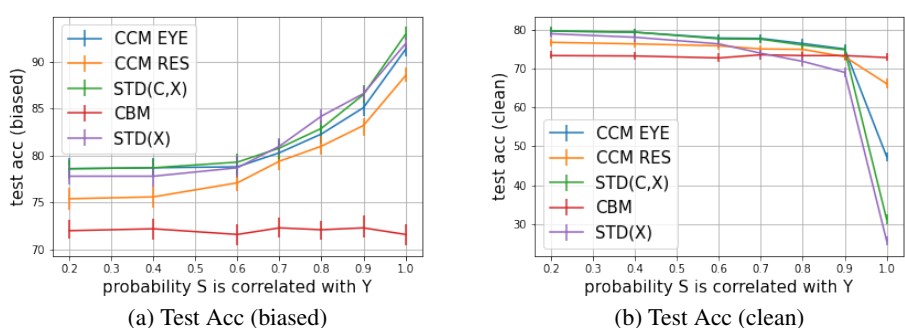

(a) Test Acc (biased)  (b) Test Acc (clean)

Figure 5: Results of relaxing **A2** by making $S$ more informative. Here, instead of generating $S$ from CBM, we correlate $S$ with $Y$ directly and sweep the value of $T$. This experiment demonstrates what happens when $S$ contains information beyond $C$ and $U$.

The choice of $T$ determines how biased the training dataset is (*i.e.*, the correlation between $S$ and $Y$), with $T = 1$ being the most biased and $T = 0$ being not biased. To ensure that the shortcut is easy to learn, we keep $T = 1$ for all experiments except when we explore CCM's sensitivity to assumptions, described later.

### A.4 Additional CUB results

**Q**: Test accuracy of CUB experiments on the biased dataset?

In the main text, we showed that CCM methods perform well when shortcuts are violated. Here we present the result of methods on the biased dataset when **A1** and **A2** are broken in **Figure 4a** and **Figure 4b** respectively. Overall, CBM approaches perform worse on the biased dataset because it lacks the ability to learn $U$, while all other approaches perform similarly.

**Q**: What if $S$ carries information outside of $C \cup U$?

The second way to break **A2** is to directly correlate $S$ with $Y$ on line 4 of the BIAS function. We sweep $T$ to control the correlation between $S$ and $Y$. As shown in **Figure 5**, as $T$ increases, the performance on the biased test set increase as well, but not the clean dataset performance, confirming that $S$ contains information outside of $C$ and $U$. We also observe that CCM RES performs worse than the standard model on both the biased and portions of the clean dataset. In contrast, CCM EYE is consistently better than $STD(X)$ on the clean dataset and comparable to it on the biased

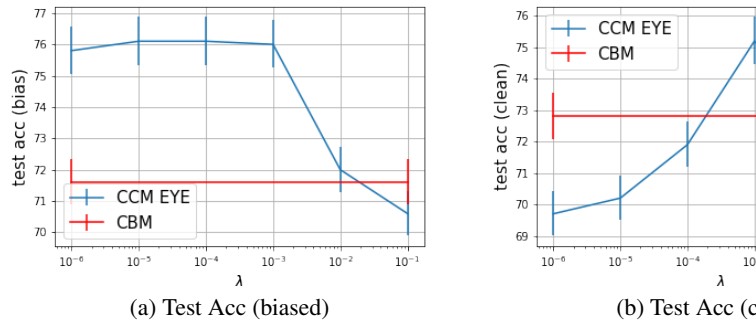

(a) Test Acc (biased)        (b) Test Acc (clean)

Figure 6: Results of sweeping $\lambda$. Without sacrificing test accuracy on the biased dataset ($\lambda \leq 10^{-4}$ in this case for the CUB dataset), increasing $\lambda$ boosts performance on the clean test set, justifying our choice of hyperparameter for CCM EYE.

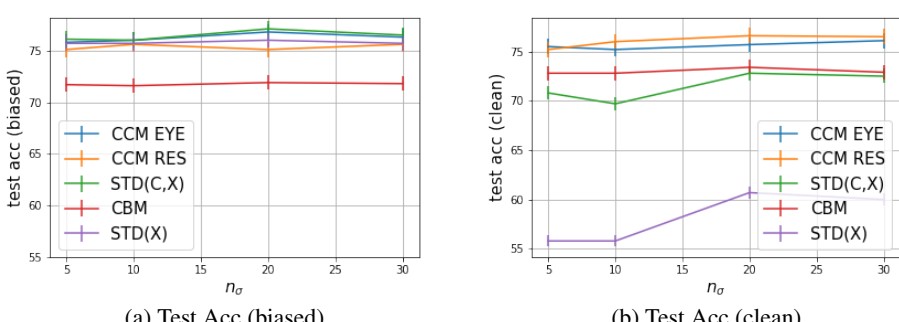

(a) Test Acc (biased)        (b) Test Acc (clean)

Figure 7: Results of sweeping number of noises ($n_\sigma$). Regardless of $n_\sigma$, CCM EYE and CCM RES outperform baselines on the clean dataset, while maintaining similar performance on the biased dataset.

dataset. Moreover, $STD(C, X)$ is comparable to CCM EYE until the shortcut is the strongest (*i.e.*, $T = 1$). This makes sense because when shortcuts are weak compared to $C$, they are not taken by $STD(C, X)$.

**Q**: How does $\lambda$ affect model performance?

Since CCM EYE has an additional parameter $\lambda$, we want to understand its effect on model performance. **Figure 6** summarizes the results on the CUB dataset. Fixing test accuracy on the biased dataset (*e.g.*, $\lambda \leq 10^{-4}$), increasing the EYE penalty monotonically increases model performance on the clean dataset. This means that credible model's principle (*i.e.*, increasing alignment with expert without sacrificing performance) could help mitigate the use of shortcut when **A1** and **A2** hold.

**Q**: How does CCM responds to different levels of shortcut?

**Figure 7** shows that regardless of $n_\sigma$, CCM outperforms baselines.

## A.5 Additional MIMIC results

We include more results of the MIMIC dataset in **Figure A.5**, **Figure A.5**, and **Figure A.5**. Each plot varies the training distribution (noted by the black line) used to train the models and compares their results on different test distributions. Note that CCM EYE consistently outperforms baselines models when the training and testing distribution are close. It only performs worse against CBM when the testing distribution is very different from the training.

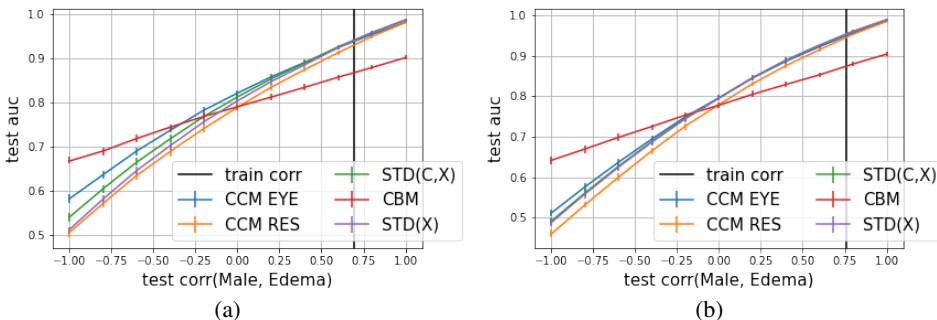

(a)              (b)

Figure 8: Result of the MIMIC-CXR experiment for different training distributions (correlations between male and edema are 0.7 and 0.76 respectively). CCM EYE consistently outperforms baselines models when the training and testing distribution are close. It only performs worse against CBM when the testing distribution is very different from the training.

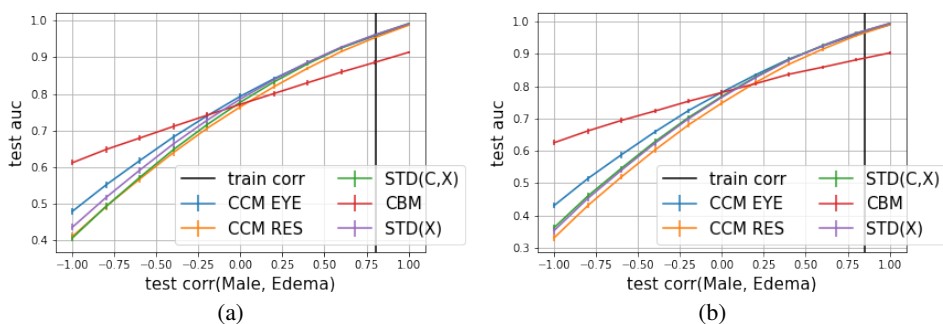

(a)              (b)

Figure 9: Result of the MIMIC-CXR experiment for different training distributions (correlations between male and edema are 0.81 and 0.85 respectively). CCM EYE consistently outperforms baselines models when the training and testing distribution are close. It only performs worse against CBM when the testing distribution is very different from the training.

### A.6 Experiments on the Physionet Challenge dataset

The **Physionet Challenge 2012** dataset [38] is a publicly available benchmark dataset from Gold-berger et al. [39] in which one aims to predict in-hospital mortality using electronic health record collected in intensive care units for $4,000$ patients. Our preprocessing follows Wang et al. [15], obtaining a feature set of size 130. In addition to the features, we have 15 variables corresponding to the Simplified Acute Physiology Score (SAPS-I) that are developed by physicians to predict ICU mortality in patients greater than the age of 15 [40]. We use those features along with age as $C$. This mimics setting where the true concepts are learned based on medical knowledge.

Here, we define shortcut variables to be variables correlated with the 15 SAPS-I variables and age. In other words, $S$ is composed of all non $C$ features that have a correlation with features in $C$ above a certain threshold. Other features are regarded as $U$ as their value is not causally related to the shortcuts. This setup mimics the setting in which shortcuts are correlated with known risk factors, motivated in Wang et al. [15].

**Model Training**: Following Wang et al. [15], we train linear models on this dataset with the Adam optimizer [41]. We randomly reserve $25\%$ of patients as the test set. Of the remaining data, we randomly split $25\%$ for validation and the rest for training. We train baseline models as well as our models using the full set of features and duplicate features in $C$ for $STD(C,X)$ to increase its chance to use $C$.

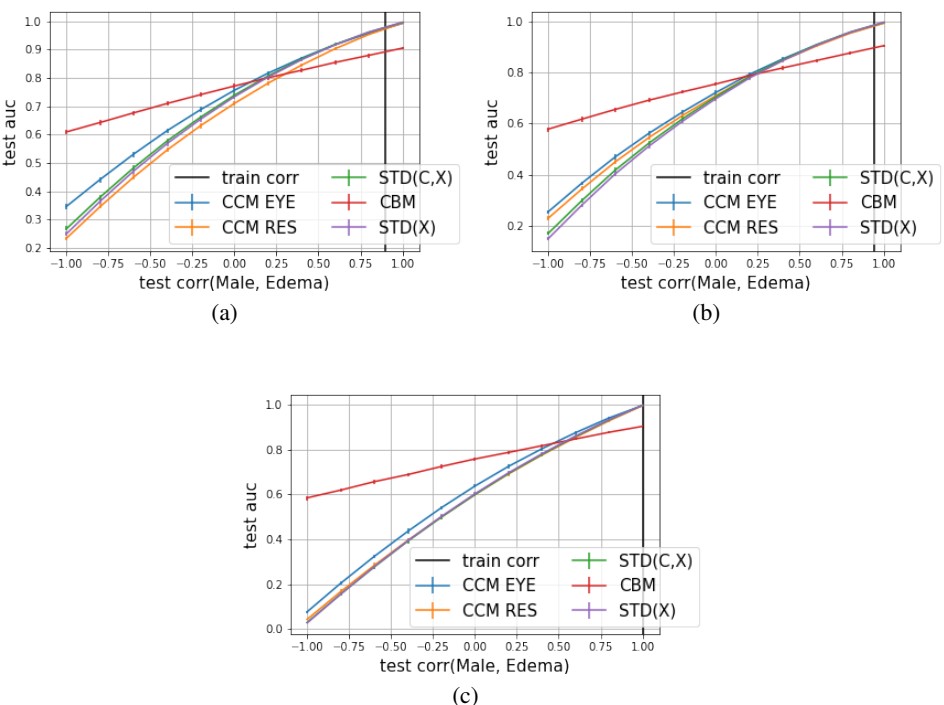

Figure 10: Result of the MIMIC-CXR experiment for different training distributions (correlations between male and edema are 0.9, 0.95, and 1 respectively). CCM EYE consistently outperforms baselines models when the training and testing distribution are close. It only performs worse against CBM when the testing distribution is very different from the training.

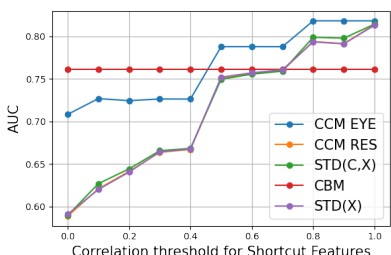

Figure 11: Treating features correlated with $C$ as shortcuts in the Physionet Challenge 2012 dataset, we measure the performance when shortcuts break (set to 0). As expected, when shortcuts are highly correlated with $C$, CCM EYE outperforms all baselines. Even when shortcuts are not highly correlated with $C$ (violating **A2**), CCM EYE is only second to CBM. In contrast, CCM RES has trouble beating the baselines because $U$ is correlated with $S$.

**Evaluation** We test model performance by setting the value for shortcut variables to 0, making them uninformative at test time. If a model is robust to $S$, this change shouldn't affect its prediction accuracy. The biased dataset performance is reported when no shortcuts are "zeroed out". This happens with a correlation threshold of 1 as no features other than $C$ have a perfect correlation with features in $C$.

**Result**: CCM EYE outperforms all other baselines in **Figure 11**. CCM EYE does not use features that are highly correlated with $C$, thus eliminating the use of $S$. In contrast, CCM RES does not do as well even when $C$ and $S$ are highly correlated (**A2**). This happens because $U$ is correlated with $S$ (yet not causally related). For example, if we fix the correlation threshold at 0.8 for shortcuts, 57% of variables in $U$ have at least 0.1 correlation with a variable in $S$.

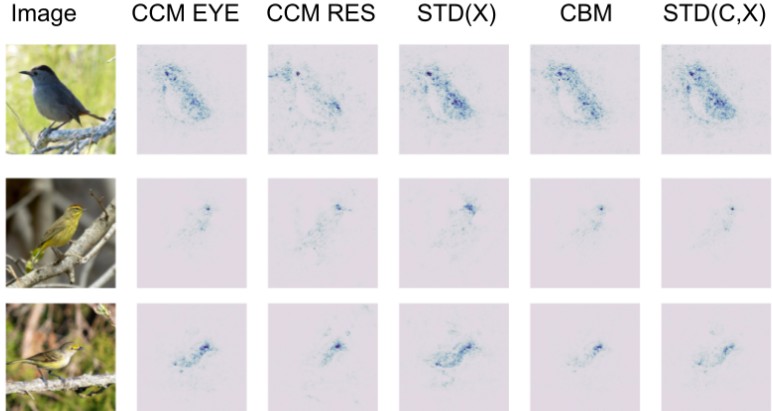

Figure 12: Saliency maps for 3 randomly selected images from the CUB dataset. In the first column, we show the original images from the dataset. Along the row following the image, we show the saliency map produced using integrated gradient [42] for our methods and baselines. Note that the visualizations are ineffective at highlighting the shortcut for this task (*i.e.*, noise), even though baseline methods are very senstive to the degree of noise as shown in **Table 1**. It is also hard to tell the difference in attribution across methods as they all correctly concentrate on the bird and do not focus on the background. This result shows that visual grounding/feature attribution can be ineffective at resolving shortcuts. We need techniques like CCMs to make robust predictions.

### A.7 Input feature attribution cannot identify shortcuts in our settings

Identifying shortcuts in images can be hard, especially when they are not tied to a salient location in an image. For example, in our settings for the bird and the chest x-ray datasets, it is unclear how to pinpoint noise or the patient sex on an image (for the same reason that it is hard to specify important regions in an image to focus on). Indeed, when we visualize where models are focusing on for both the CUB and the MIMIC-CXR dataset in **Figure 12** and **Figure 13** respectively on randomly chosen images, the visualization does not highlight the shortcuts. In fact, it appears that the least robust model, $STD(X)$, has similar attribution compared to the other models. Here the attributions on image pixels are obtained by summing the absolute value of integrated gradient [42] (a popular feature attribution method) across the image's RGB channels.

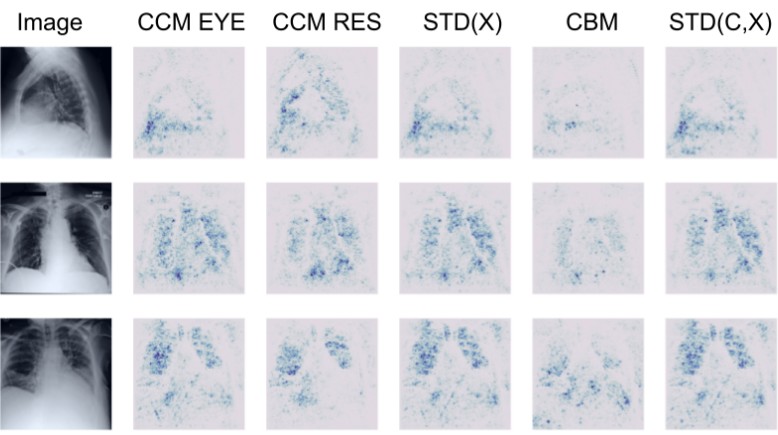

Figure 13: Saliency maps for 3 randomly selected images from the MIMIC-CXR dataset. In the first column, we show the original images from the dataset. Along the row following the image, we show the saliency map produced using integrated gradient [42] for our methods and baselines. Note that the visualizations are ineffective at highlighting the shortcut for this task (*i.e.*, patient sex), even though baseline methods are very senstive to the sex of a patient as shown in **Figure 3**. This result shows that visual grounding/feature attribution can be ineffective at resolving shortcuts. We need techniques like CCMs to make robust predictions.