# OpenReview forum: "Learning Concept Credible Models for Mitigating Shortcuts"
_NeurIPS.cc/2022/Conference — NeurIPS 2022 Accept_

### Official Review · Reviewer_UnPr · 2022-07-05

**Rating:** 7
**Confidence:** 3
**Soundness:** 3 good
**Presentation:** 4 excellent
**Contribution:** 3 good

**Summary:**

The paper proposes concept credible models (CCMs) to mitigate spurious correlations/shortcuts $S$ by assuming access to some known concepts ($C$) (e.g., attribute annotations/transferred features) and then learning remaining unknown concepts ($U$). It first proposes: CCM RES, which fits a residual model to account for the unknown concepts relevant for the task. To account for the case when shortcuts are correlated with $U$, it further proposes CCM EYE which under some assumptions, mitigates reliance on $S$, even when they are correlated with $U$.

**Questions:**

Please address the points mentioned in the weaknesses section.
With attribute-based CBMs, the predictions are grounded on semantically meaningful factors of the data. But, using transferred features for $C$ and with residual features $U$, the implicitly learned features likely do not have this property. Could you elaborate more on the implications and the potential ways to close the interpretability gap?


**Limitations:**

I think the section on ethical considerations is adequate.

**Strengths And Weaknesses:**

$\textbf{Strengths}$


1. This work addresses an important weakness of the original Concept Bottleneck Model (CBM), which assumes that all the concepts required for the task are known in advance and the known set can be used to perform the task. CBM, in its original formulation, is incapable of learning additional concepts and shows inferior results if other concepts are indeed necessary. The proposed CCM RES/EYE models on the other hand, are capable of learning unknown concepts too, while still remaining robust to the shortcuts, so this has practical value.

2. Leveraging domain knowledge (e.g., extra labels or features that represent task-relevant concepts) is a realistic way of using deep learning models robustly in practice. Robustness can be increased by relying on those task-relevant features instead of the shortcuts. One major advantage of the proposed approach is that this is done without actually accessing information about the shortcuts. This is different from many existing works which use information about spurious factors e.g., to divide the dataset into multiple environment (e.g., as done in IRM).

3. CCM EYE/CCM RES obtain comparable accuracies on both biased and clean test accuracies for CUB, showing that it is relying on the core features instead of the spurious ones. Adaptation of EYE regularization on the concept space improves alignment with extra information to mitigate shortcuts. This is logically sound.

4. Both assumptions $A1$ and $A2$ seem reasonable even for practical scenarios. I like the studies performed when the assumptions are violated. When violating $A1$ ($C$ is counterfactually invariant to $S$), the method remains robust, unless $C$ is heavily impacted by $S$. Violation of $A2$ ($S$ is redundant given $C$) shows that it is not necessary to have perfect $C$ i.e., not all dimensions need to be relevant leaving room for even the experts to be imperfect.


$\textbf{Weaknesses}$

1. I have a question in regards to model interpretability. With attribute-based CBMs, the predictions are grounded on semantically meaningful factors of the data. But, using transferred features for $C$ and with residual features $U$, the implicitly learned features likely do not have this property. Could you elaborate more on the implications and the potential ways to close the interpretability gap?

2. While the method compares accuracies on biased/clean test sets to provide evidence that the model is relying on concepts instead of shortcuts, the study does not provide qualitative visualizations/quantitative analyses to confirm that. I wish it was studied on other datasets where one could perform visual grounding analysis (compute segmentation metrics) to show it is not looking at the spurious regions of images.

3. It would have been impactful to test on WILDS [1] and other more realistic benchmarks instead of introducing artificial shortcuts in CUB. The distribution shift/shortcuts should be reflective of the real world challenges. Furthermore, it would also be interesting to see how one could inject known concepts/domain expertise for such cases.

[1] Koh, Pang Wei, et al. "Wilds: A benchmark of in-the-wild distribution shifts." International Conference on Machine Learning. PMLR, 2021.

Missing reference:
Learning additional concepts for CBMs was explored in [2], so it can be cited.

[2] Sawada, Yoshihide, and Keigo Nakamura. "Concept Bottleneck Model With Additional Unsupervised Concepts." IEEE Access 10 (2022): 41758-41765.

---

> ### Author Response · Authors · 2022-08-02
> **Complementing CCM with concept level interpretation methods, one can interpret and validate U to further mitigate shortcut learning.**
>
> Thank you for carefully reading the paper and for your thoughtful feedback.
>
> **1. What are the implications and potential ways to close the interpretability gap between CBM and CCM?**
>
> The question regarding grounding implicitly learned features (U) is very relevant and important. Indeed, while CCM is capable of ruling out shortcuts that are redundant given C, it does not replace the need to carefully interpret and examine concepts picked up by the model (i.e., U), as some shortcuts contain more information given C. This requires examining what U encodes and letting domain experts validate the learned concepts. Fortunately, this is an active area of research and there are already ways to close the interpretation gap (i.e., interpret learned concepts), both supervised [1] and unsupervised [2]. They work by defining a set of concepts to test. For each test concept, they learn a concept vector and compare it to the activation in U. If similar, U is said to encode the test concept. Furthermore, if the encoded concept is deemed important for the prediction (e.g., large directional derivative of the model output on the concept vector), the test concept provides a grounding for U. Complementing CCM with the interpretation of U, is an interesting future direction that could work to further mitigate shortcut learning.
>
> **2. Visual grounding analysis.**
>
> We like the suggestion to perform visual checks. We will include qualitative feature attribution results comparing CCMs and baselines in the final version.
>
> **3. Testing on WILDS and other more realistic datasets.**
>
> While testing on WILDS will further strengthen the paper, defining prior knowledge for these tasks is non-trivial. We focused on clinical tasks using real-world data since we had access to clinical expertise to help design and validate the experimental setup. We believe that our current experiments have already demonstrated the effectiveness of CCMs and leave further investigation to applications with domain experts.
>
> Thank you again for your thorough review. We’ll incorporate the missing citation into the paper.
>
> [1] Been Kim, Martin Wattenberg, Justin Gilmer, Carrie Cai, James Wexler, Fernanda Viegas, et al. Interpretability beyond feature attribution: Quantitative testing with concept activation vectors (tcav). In ICML, 2018.
>
> [2] Yeh, Chih-Kuan, et al. "On completeness-aware concept-based explanations in deep neural networks." Advances in Neural Information Processing Systems 33 (2020): 20554-20565.

---

> > ### Comment · Reviewer_UnPr · 2022-08-05
> > **Response to the Authors**
> >
> > The authors have adequately answered my questions. I think visual checks will be a valuable addition. And yes, while more datasets would have helped, I do not think they are absolutely necessary for acceptance. So, I keep my score intact.

---

### Official Review · Reviewer_LUM8 · 2022-07-11

**Rating:** 6
**Confidence:** 3
**Soundness:** 3 good
**Presentation:** 3 good
**Contribution:** 2 fair

**Summary:**

The authors proposed a new method for mitigating shortcuts. They built upon Concept Bottleneck Models (CBM) to also learn the unknown concepts. The proposed approach is simple and easy to understand, with the detailed analysis presented for linear models. Experiments on the CUB dataset show that the method is able to mitigate the shortcuts.


**Questions:**

1. Why does CBM perform worse than STD(X) in figure 3. for T=0?


**Limitations:**

Yes, the authors have discussed the limitations and potential negative societal impact.

**Strengths And Weaknesses:**

### **Stregnths**
1. The paper is well-written and easy to follow. The paper attempts to solve an important problem, i.e., mitigating shortcuts.
2. The method tries to solve the limitations of CBMs and use both known and unknown concepts for learning.
3. Detailed analysis of the method and assumption is presented for the linear model.
4. Experiments on CUB dataset show clear performance gains.

### **Weakness**
1. Experiment on the MIMIC-CXR dataset doesn’t show significant performance gains.
2. Only two experiments are included in the paper. More experiments should be conducted (CelebA, etc.).
3. Analysis presented in section 3 is for linear models. It would be interesting to see a similar analysis for non-linear models.

---

> ### Author Response · Authors · 2022-08-02
> **STD(X) can perform better than CBM when STD(X) learns robust features.**
>
> Thank you for your encouraging feedback. We are glad that you appreciate the problem setup. Below, we first answer your question and then address the weaknesses you identified.
>
> **1. Why does CBM perform worse than STD(X) in figure 3 for T=0?**
>
> In Figure 3, regardless of the test correlation, CBM could be worse than STD(X). Recall that we expect C to be robust against shortcuts. Therefore, CBM should be less sensitive to the change in distribution caused by shortcuts, resulting in a flatter line in Figure 3 (which we observe).  However, there is no guarantee that CBM’s performance will be good with no correlation between the task and the shortcut. Consider the extreme case that C is an empty set. CBM’s performance will be flat but also low. STD(X), on the other hand, is likely to depend on shortcuts but it may also learn features (e.g., U) that are robust against shortcuts, therefore yielding decent performance even with no correlation between the task and shortcut.
>
> **2. Experiment on the MIMIC-CXR dataset doesn’t show significant performance gains.**
>
> The MIMIC-CXR experiment shows that CCM is more robust to shortcuts than standard models and is much better than CBM under mild shortcut distribution shift (Figure 3). When the test correlation is negative, CCM outperforms STD, and when the correlation is positive it outperforms CBM. Overall, CCM’s performance is either significantly better or equal than all other baselines regardless of distribution shift.
>
> **3. Should more experiments be conducted?**
>
> The supplemental material contains additional experiments on the Physionet 2012 challenge dataset. While we agree that more datasets can help better validate our approach, we also want to highlight that for each dataset, we conducted detailed analysis on what happens when the assumptions fail. Despite being limited to 3 datasets, the number of experiments should be sufficient to evaluate our approach.
>
> **4. Extending analysis to non-linear models**
>
> Extending our analysis to nonlinear models is indeed important. However, this is non-trivial and is best tackled in future work. In this paper, we seek to lay the foundation of CCM and focus on its empirical evaluation.
>
> Again, thank you for the valuable feedback.

---

> > ### Comment · Reviewer_LUM8 · 2022-08-08
> > **Response to the author rebuttal**
> >
> > Thank you for the response. I have revised my rating. This work will be useful for the spurious correlation/shortcuts community.

---

### Official Review · Reviewer_HAYL · 2022-07-11

**Rating:** 5
**Confidence:** 4
**Soundness:** 3 good
**Presentation:** 3 good
**Contribution:** 2 fair

**Summary:**

This paper suggests learning unknown concepts besides the known ones (prior knowledge) in other to better tackle shortcut learning.  The authors have tested the method on the CUB (birds) dataset and edema prediction from x-ray images.

**Questions:**

I would like the authors to define what U refers to in practice, particularly in a clinical setting.

**Limitations:**

The limitation of the work can be discussed in more detail in the conclusion section.

**Strengths And Weaknesses:**

Pros:

- This paper is in the direction of incorporating prior knowledge into the machine learning models which is important for addressing various problems including mitigating shortcut learning.
- Having access to domain knowledge is not easy/cheap but I agree that without prior knowledge it is difficult to mitigate spurious correlations
- The writing is clear and flows well.
- The linear analysis is interesting.


Cons:

- The method's novelty is in forcing the learner to learn unknown (U) concepts in addition to known ones (C). I'm not sure what U stands for in practice or why we need it. Could the authors provide specific real-world examples? Consider C as the known concept in a simple classification setup, such as how a cat looks in terms of body shape, texture/colour of skin, facial attributes, and so on. So, if we encourage the model to learn these features, what is U then? Do you mean the C's that we missed for some reason? For example, C* = {C,U} and we want to learn C*. If so, learning those should be difficult, especially in clinical settings where one must constantly check whether the newly learned features (U) make sense to include in the final prediction. Given an x-ray, if a doctor knows that a crack (the known concept, C) in a bone indicates that the bone is broken, what else do we need to learn as U?

- How strong is the added noise to the samples in the CUB experiment? Shouldn't training and selecting checkpoints on noisy data but testing on clean data result in poor performance if the model hasn't seen any clean images during training?

- Transfer learning appears to be unusual for learning concept features when clear connections between the proxy and man tasks are difficult to establish. Could the authors elaborate on the reasoning behind the transfer learning in experiment 4.2 in the paper, i.e., can they confirm that such an experiment makes clinical sense? Did the authors discuss with clinicians the possibility that training a model on cardiomegaly (the proxy task) would teach it the concept features C on diagnosing edema (the main task)?

- I recommend that the authors remove experiment 4.2 from the main paper if they are unable to confirm (with clinicians) that the transfer learning component makes sense (cardiomegaly —> edema). The authors can substitute the Physionet Challenge experiment in the appendix for this experiment.

---

> ### Author Response · Authors · 2022-08-02
> **Our experiments were reviewed by a clinician. U in a clinical setting can refer to risk factors yet to be discovered or risk factors not specified in training.**
>
> Thank you for your thoughtful feedback. We are glad that you too find our problem important.
>
> **1. What does U stand for in practice, why do we need it?**
>
> Your understanding about U is correct: “C* = {C,U} and we want to learn C*”. Upon reflection, we agree that the example of an X-ray with a bone fracture is not the best example, since in this simple task C is enough (e.g., C* = {C}). However, there are other clinical settings in which clinicians have some prior knowledge on what is relevant, but do not know everything that is relevant. E.g., when learning to diagnose the etiology of acute respiratory failure (ARF), clinicians are only accurate about 70% of the time. Recently ML models have been shown to improve on this accuracy and have also been shown to generalize across hospitals, suggesting that ML models are using additional features beyond those that humans rely on. To further elaborate on this point, quoting from a clinical expert,
> “we need U because experts have lots of knowledge about what might be relevant to a specific clinical setting, but we certainly don’t know everything, or may not be able to always appreciate what is exactly relevant in each setting. For example, in this recent JAMIA publication on ARF [1], the model learned that a ‘saber sheath trachea’ was relevant for identifying patients with a COPD exacerbation. If you would have asked me to come up with the set of features a model should use to identify COPD, I would have said things like ‘lung hyperinflation’, but ‘saber sheath trachea’ is not something I would have appreciated as important. Although if you dig around, you can find papers that have shown this [2]: it turns out, saber sheath trachea was important in this study. But more generally, there are likely findings out there that have not previously been recognized as important—I can’t really provide an example of something I don’t yet know, the point is that we are making medical discoveries all the time, we don’t know everything.”
> Even when all relevant risk factors are known a priori, sometimes the universe of possibly relevant risk factors is large, such that collecting and labeling all these features for a particular problem is unrealistic. In such cases, the flexibility yielded by including U helps. In some cases, U may turn out to be something known before, e.g. Saber Sheath trachea, whereas other times it might be a brand new finding.
>
> **2. Learning U is difficult.**
>
> You brought up a great point: learning U is hard. Indeed, U needs to be rigorously validated before being put into use. This is a future direction one could pursue. E.g., one may use concept-level model interpretation methods to gain a better understanding of U and have clinicians weigh in on the validity. However, even without inspecting U, we showed that CCM mitigates shortcuts better than a standard model, a significant contribution in and of itself.
>
> **3. Does transfer learning in 4.2 make sense?**
>
> We think it does because directly expressing concepts on the input space is difficult. In contrast, having an expert provide related tasks is more natural. Indeed, a clinician did review this setup and suggested (cardiomegaly —> edema).
>
> **4. Noise strength in the CUB experiment?**
>
> Details on noise level are included in supplemental material A.3. Here, the maximum variance on noise is 0.1. To improve clarity, we can move this to the main paper.
>
> **5. Shouldn't training and selecting checkpoints on noisy data but testing on clean data result in poor performance if the model hasn't seen any clean images during training?**
>
> In Table 1, we see that the results indeed degrade for the standard models. For other models, the difference is not significant because they do not rely to the same extent on shortcuts (noise level), in which case the covariate shift (due to noise) does not have the same negative effect on performance.
>
> Again, thank you for your constructive feedback. In the camera-ready version, we will update the motivating example to be one in which including U helps as described above.
>
> [1] https://academic.oup.com/jamia/article/29/6/1060/6546605
>
> [2] https://link.springer.com/article/10.1007/s11547-013-0318-3

---

### Meta-Review · Area_Chair_RiRk · 2022-08-31

**Recommendation:** Accept
**Confidence:** Certain

**Metareview:**

The paper considers the important problem of shortcut learning and the existing solutions and suggests learning unknown concepts besides the known ones (prior knowledge) in other to better tackle shortcut learning. They built upon Concept Bottleneck Models (CBM) to also learn the unknown concepts. The proposed approach is simple and easy to understand, with the detailed analysis presented for linear models.The authors have tested the method on the CUB (birds) dataset and edema prediction from x-ray images.
The considered problem is important and the experiment results validate the effectiveness of the proposed approach. Moreover, the authors were able to address reviewers questions and concerns during the rebuttal period. Therefore, I suggest the paper to get accepted.

**Award:**

No

---

### Decision · Program_Chairs · 2022-09-14

Accept